# Unraveling Phylogenetic Relationships Among Six *Miscanthus* Andersson (Poaceae) Species Through Chloroplast Genome Analysis

**DOI:** 10.3390/genes16101175

**Published:** 2025-10-10

**Authors:** Ji Eun Kim, Yang Su Kim, Gyu Young Chung, Hyeok Jae Choi, Chang-Gee Jang, Hoe Jin Kim, Chae Sun Na

**Affiliations:** 1Wild Plant Seed Division, Baekdudaegan National Arboretum, Bonghwa 36209, Republic of Korea; 2Department of General Affairs, Gangneung-Wonju National University, Gangneung 25457, Republic of Korea; 3Division of Horticulture and Medicinal Plant, Andong National University, Andong 36729, Republic of Korea; 4Department of Biology and Chemistry, Changwon National University, Changwon 51140, Republic of Korea; 5Department of Biology Education, Kongju University, Gongju 32588, Republic of Korea; 6Global Seed Vault Center, Korea Arboreta and Gardens Institute, Sejong 30129, Republic of Korea

**Keywords:** *Miscanthus*, phylogenetic relationship, chloroplast genome

## Abstract

Background/Objectives: *Miscanthus* Andersson, a genus of perennial grasses that includes wild relatives of key crop species, remains poorly characterized in terms of genetic diversity and evolutionary relationships. The aim of this study was to elucidate the phylogenetic structure of *Miscanthus* through comparative genomic analysis of the chloroplast genomes of six Korean species. Methods: Complete chloroplast genomes were assembled and analyzed for six *Miscanthus* species. Informative nucleotide motifs and their associated gene locations were identified as potential markers, and their phylogenetic relationships with related crops were examined. Results: The chloroplast genomes exhibited a conserved quadripartite structure, with genome sizes and GC contents within typical ranges. Analysis of codon usage showed a preference for A/U-ending codons, consistent with patterns in other angiosperms. Simple sequence repeats and long repeats demonstrated non-random distributions, indicating their value as molecular markers for phylogenetic and population studies. Comparative analyses confirmed structural conservation across *Miscanthus* species, whereas variation in non-coding regions provided important phylogenetic signals. Phylogenetic reconstruction based on 21 chloroplast genomes revealed four major clades, corroborating previous findings and highlighting complex evolutionary relationships within *Miscanthus*, including close affinities between African and Himalayan species and the genus *Saccharum* L. Conclusions: This study provides complete chloroplast genomes of six *Miscanthus* species, contributing to enhanced understanding of the relationships within the subtribe Saccharinae. The findings support the inclusion of *Miscanthus* species in the Korea Crop Wild Relatives inventory and highlight their potential as a genetic resource for breeding programs aimed at enhancing crop resilience to environmental stress.

## 1. Introduction

Agriculture is highly vulnerable to a range of disasters and crises, including climate change, extreme weather conditions, and pest outbreaks. These challenges continue to threaten global food security, underscoring the urgent need for effective resilience strategies, particularly in crop management, to ensure a stable and sustainable food supply that can meet the demands of a growing population [1]. In this regard, Crop Wild Relatives (CWRs)—wild species genetically related to domesticated crops—represent a critical genetic reservoir for enhancing crop traits such as resilience, yield, and adaptability under changing climatic and environmental conditions [2,3,4]. Emerging research continues to highlight the essential role of CWRs in breeding climate-resilient crop varieties [5,6,7,8]. Genetic diversity from CWRs can be introduced into crops, thereby enhancing their ability to cope with future environmental stresses.

The family Poaceae is the fifth largest group of angiosperms and the second largest group of monocots, with approximately 700 genera and between 11,000 and 12,000 species [9,10,11,12]. The genus *Miscanthus* Andersson is native to East and Southeast Asia, and is primarily distributed in temperate and subtropical regions, showing adaptability to diverse habitats and a survival ability at high altitudes [13,14,15]. Traditionally, the genus has been valued for its domestic uses, including livestock feed, green manure, and roofing material for traditional homes [15]. Furthermore, *Miscanthus* has recently attracted significant industrial attention owing to its potential in bioenergy production and its ability to enhance agricultural resilience [16,17]. Notably, within the same tribe, Andropogoneae, key crops such as maize (corn), sorghum, and sugarcane highlight the agricultural and industrial significance of this lineage. Nevertheless, the phylogenetic relationships among *Miscanthus* species remain ambiguous, limiting efforts to systematically harness their genetic diversity.

The structure of the chloroplast genome is generally well conserved across plant species, comprising a circular, double-stranded DNA molecule [18,19]. This genome is typically organized into four distinct regions: a large single-copy (LSC) region of approximately 80–90 kb, a small single-copy (SSC) region ranging from 16 to 27 kb, and a pair of inverted repeat (IR) regions, each measuring approximately 20–28 kb [20,21,22,23]. Owing to its unique characteristics, such as its predominantly maternal inheritance pattern across angiosperms, the chloroplast genome serves as a powerful tool for investigating plant evolutionary history and reconstructing phylogenetic relationships [24,25,26,27,28].

This research focused on constructing and characterizing the chloroplast genomes of *Miscanthus* species native to South Korea while also updating the Korea Crop Wild Relatives (KCWRs) inventory. Specifically, this study involved the de novo assembly and functional annotation of chloroplast genomes, followed by phylogenetic assessments to investigate both intra-genus relationships and connections with the closely allied taxa. Our findings on the chloroplast genome structure, genetic variability, and phylogenetic relationships in *Miscanthus* offer valuable information to guide the conservation and utilization of wild relatives of crops.

## 2. Materials and Methods

### 2.1. Plant Materials

Among the species of *Miscanthus* distributed in Korea [29], six species, *M. sinensis* var. *gracillimus*, *M. sinensis* f. *chejuensis*, *M. sacchariflorus*, *M. sinensis*, *M.* × *longiberbis*, and *M. sinensis*, were analyzed in the study (Figure 1, Table 1). Seeds of each species were sown, and young leaves from germinated individuals were collected for analyses. The seeds and DNA have been deposited in the Baekdudaegan National Arboretum Seed Bank and are stored in a deep freezer.

### 2.2. Methods for DNA Isolation and Chloroplast Genome Assembly

Genomic DNA was isolated from plant tissues using the NucleoSpin Plant II Kit (Macherey-Nagel, Duren, Germany) following the manufacturer’s guidelines. The integrity and concentration of the extracted DNA were evaluated using a NanoDrop 2000 spectrophotometer (Thermo, Wilmington, DE, USA) and Qubit 3.0 Fluorimeter (Invitrogen, Carlsbad, CA, USA). Library preparation for paired-end sequencing was carried out with the TruSeq Nano DNA Kit (Illumina, San Diego, CA, USA), and sequencing was performed using the NovaSeq platform. On average, 3.0–3.6 Gb of raw reads were generated per sample (19.7–23.8 million paired-end reads), resulting in more than 100× coverage of the chloroplast genome. The sequencing quality was high, with Q20 values of 97.2–97.9% and Q30 values of 92.2–94.5% across samples. Raw reads were quality-filtered by removing adapter sequences and discarding those with Phred scores below 20. The chloroplast genomes were assembled using CLC Assembly Cell v4.2.1 (CLC Inc., Aarhus, Denmark). Annotation of coding and non-coding regions was performed with GeSeq [30], and annotation errors were manually corrected using Geneious Prime v2025.1.2 (Biomatters Ltd., Auckland, New Zealand) [31]. A circular map of the chloroplast genome was visualized using OGDRAW v1.31. [32]. All resulting chloroplast genome sequences have been deposited in the GenBank databases (NCBI and K-BDS).

### 2.3. Comprehensive Genome Analysis of Miscanthus Species

The codon usage bias, expressed as Relative Synonymous Codon Usage (RSCU) values, was analyzed using protein-coding sequences in MEGA 12.0.11 [33]. RSCU values provide a measure of codon bias, where values below 1 indicate under-representation, values near 1 suggest random usage, and values above 1 reflect preferential usage. In this study, codons with RSCU < 0.6 were considered extremely underrepresented, whereas those with RSCU > 1.6 were regarded as highly preferred [34,35]. Using MISA-web [36,37] simple sequence repeats (SSRs) were screened under predefined thresholds: 10 copies for mononucleotides, 5 for dinucleotides, 4 for trinucleotides, and 3 for higher-order motifs (tetra- to hexanucleotides). Additionally, REPuter [38] was employed to search for forward (F), palindromic (P), reverse (R), and complementary (C) repeats within a length range of 20–50 bp by applying a Hamming distance threshold of 3 (https://bibiserv.cebitec.uni-bielefeld.de/reputer, accessed on 12 May 2025). In angiosperms, inverted repeats (IR) in chloroplast genomes are highly conserved and play a role in genome stability, with IR boundary expansion and contraction being key mechanisms driving genome size variation [39]. The IR region boundaries across six *Miscanthus* species were visualized using the IRscope server (https://irscope.shinyapps.io/irapp/, accessed on 14 May 2025) [40]. To identify regions of high nucleotide diversity (π) within the chloroplast genome, we employed DnaSP v5 [41], with a sliding window analysis configured at a 600 bp window size and a 200 bp step size. MAFFT v7.490 [42,43] was used to align the sequences, which were edited in Geneious Prime v2025.1.2 (https://www.geneious.com). This approach effectively facilitated the detection of conserved and variable regions. Divergence among six *Miscanthus* species was analyzed using the mVISTA tool (http://genome.lbl.gov/vista/mvista/submit.shtml, accessed on 15 May 2025) [44,45]. Using the LAGAN algorithm [46] sequences were aligned against the reference chloroplast genome of *Saccharum officinarum* (GenBank accession Number: MZ328080).

### 2.4. Phylogenetic Relationships Among Miscanthus Species

Chloroplast genomes of six *Miscanthus* species were analyzed, alongside five *Saccharum* species and two *Zea* species (included as outgroups), to clarify phylogenetic relationships within *Miscanthus*. The genomic data were sourced from NCBI GenBank (https://www.ncbi.nlm.nih.gov/, accessed on 26 January 2025). Species categorization (including CWRs, landraces, and cultivated types) was based on GRIN-Global [47]. Using PhyloSuite v1.2.3 [48], we obtained the protein-coding gene sequences required for further analysis. Sequence alignments were generated using MAFFT v7.490 [42,43] and further analyzed using Geneious Prime v2025.1.2 (https://www.geneious.com). A concatenated alignment of all protein-coding genes was constructed and used to generate a phylogenetic tree with the ML method in IQ-TREE [49]. The software selected K3Pu+F+I as the best-fitting model, and node reliability was tested with 1000 ultrafast bootstrap replicates.

## 3. Results

### 3.1. Structural and Genomic Characteristics of the Chloroplast Genomes

The complete chloroplast genomes of the six *Miscanthus* species displayed high levels of conservation in terms of gene content, overall genome length, and GC composition (Figure 2, Table 2). The total genome sizes ranged from 141,313 bp to 141,377 bp. They possessed a quadripartite structure, consisting of LSC regions (83,116–83,182 bp), IR regions (22,798–22,799 bp), and SSC regions (12,557–12,659 bp). The GC content of all chloroplast genomes was 38.4%. The consistent GC content across all species reflects the generally AT-rich composition characteristic of chloroplast genomes. A noticeable increase in GC content, ranging from 43.9% to 44.1%, was observed in the IR region, contrasting with the lower GC contents in the LSC (36.2%) and SSC (32.7–32.8%). This pattern suggests greater stability in the IR region [50].

In total, 135 genes were annotated in each genome, comprising 89 protein-coding genes, 38 tRNA genes, and 8 rRNA genes (Table 2 and Table 3). The genes within the chloroplast genome were divided into 18 functional categories. Among these, 10 protein-coding genes (*ycf73*, *ycf2*, *ycf15*, *rps7*, *rps19*, *rps15*, *rps12*, *rpl2m rpl23*, and *ndhB*), 8 tRNA genes (*trnA^UGC^*, *trnH^GUG^*, *trnI^GAU^*, *trnL^CAA^*, *trnN^GUU^*, *trnR^ACG^*, *trnI^CAU^*, and *trnV^GAC^*), and 4 rRNA genes (*rrn16*, *rrn23*, *rrn4.5*, and *rrn5*) were duplicated in the IR regions. The *rps12* gene was trans–spliced, with its 5′ exon positioned in the LSC region and its 3′ exon situated in the IR region. The *matK* gene was located within the intron of *trnK*^UUU^.

### 3.2. Analysis of Relative Synonymous Codon Usage (RSCU) Patterns

For each species, RSCU values were derived based on the chloroplast genome’s protein-coding genes, which included 64 codons representing 21 amino acids (Figure 3). The total codon counts were 18,959 for *M. sinensis* var. *gracillimus*, *M. sinensis* f. *chejuensis*, and *M. sinensis*; 18,960 for *M. sacchariflorus*, and *M.* × *longiberbis*; and 18,981 for *M. sinensis* var. *purpurascens*. Leucine (Leu), encoded by UUA, UUG, CUU, CUC, CUA, and CUG, was the most abundant amino acid, with 2071–2078 codons (10.87–10.89%). In the six species analyzed, 32 codons had RSCU values higher than 1, and another 32 codons had values lower than 1. The UUA codon had the highest RSCU value at 2.03, whereas the CUG codon had the lowest, ranging from 0.32 to 0.33. Both AUG (Met) and UGC (Trp) displayed an RSCU value of 1, indicating unbiased codon usage. Codons with A or U as the terminal base (GCU, AGA, UUA, UCU, and ACU) exhibited RSCU values exceeding 1.6, indicating high bias. Conversely, 22 codons (GCG, CGC, CGG, AGG, AAC, GAC, UGC, CAG, GAG, GGC, CAC, AUC, CUC, CUG, AAG, CCG, UCG, AGC, ACG, UAC, GUC, and GUG) displayed markedly low biases, with RSCU values under 0.6.

### 3.3. Analysis of SSR Motifs and Large Repeats

The presence and distribution of SSRs were analyzed within the chloroplast genomes of six *Miscanthus* species (Figure 4). Mononucleotide repeats, accounting for 60.5–66.7% of the SSR motifs, ranged from 26 to 31 in number. In total, 5 di-, 1 tri-, 9 tetra-, 1 penta-, and 1 hexanucleotide repeats were found. Notably, the hexanucleotide motif was absent only in *M. sinensis* var. *purpurascens* (Figure 4A). In total, 43–48 SSRs were identified, with the lowest number (43) recorded in *M.* × *longiberbis* and the highest (48) in *M. sinensis* var. *gracillimus* and *M. sinensis* f. *chejuensis*. The LSC region exhibited the highest SSR frequency, containing 33–38 SSRs, followed by the SSC region, which contributed 6 SSRs. The IRa and IRb regions displayed relatively low and consistent SSR counts, with 2 SSRs each (Figure 4B). The SSR motifs predominantly consisted of A/T repeats, with the majority of motifs exclusively composed of these bases (Figure 4C).

Analysis of long repeat sequences indicated that forward and palindromic repeats were more frequently observed than reverse repeats, whereas no complementary repeats were identified across the six *Miscanthus* species (Figure 5). Among these long-repeat sequences, the majority were concentrated within the 30–39 bp range, accounting for 59.2% of the total, with 27–29 repeats identified in this size category (Figure 5A). In total, 49 long-repeat sequences were identified across the complete chloroplast genomes of six *Miscanthus* species, comprising 34–37 forward (F), 10–12 palindromic (P), and 3–5 reverse (R) repeats. Notably, reverse repeats were absent from the genomes of *M. sinensis*, *M. sinensis* f. *chejuensis*, and *M. sinensis* var. *gracillimus* (Figure 5B). Further analyses of these repeats could facilitate the development of molecular markers for species identification.

### 3.4. Comparison of Chloroplast Genome Sequences and Assessment of Nucleotide Diversity

IR regions may expand or contract over time, leading to structural alterations in the chloroplast genome, including changes in gene copy number and the formation of pseudogenes at the junction boundaries (Figure 6). These structural variants are often considered valuable markers in evolutionary studies [51,52]. IRscope was employed to investigate the boundary regions and adjacent genes of the chloroplast genomes in six *Miscanthus* species, with particular attention to the contraction of junction sites. The IR regions among *Miscanthus* species displayed a high degree of conservation, with lengths ranging narrowly from 22,798 to 22,799 bp. This minimal variation suggests structural stability of the IR regions across the genus. Boundary analysis of the chloroplast genomes revealed that the genes *rpl22*, *rps19*, *ndhF*, *ndhH*, *rps15*, and *psbA* were consistently positioned at the four junction sites: LSC–IRb (JLB), SSC–IRb (JSB), SSC–IRa (JSA), and LSC–IRa (JLA). Notably, ndhF was located across the SSC–IRb boundary. The conserved gene arrangement at these junctions, together with the uniform IR length, reflects the overall structural integrity of the chloroplast genomes within the *Miscanthus* genus.

Several highly variable loci were detected through nucleotide diversity analysis, offering valuable targets for species-level discrimination and evolutionary studies (Figure 7). Nucleotide diversity (π) values within the six *Miscanthus* species ranged from 0 to 0.015. The analysis primarily focused on intronic regions, where the majority of the observed variability was concentrated. In the LSC and SSC regions, three chloroplast gene regions exhibited high diversity (π > 0.006), comprising *trnC*^GCA^, *trnL*^UUA^–*trnF*^GAA^, and *trnL*^UAG^–*ccsA*. Conversely, no significant diversity (π > 0.006) was detected in the IR regions.

To assess DNA sequence variability in these species relative to those in other crops, we performed a comparative analysis of the chloroplast genomes from six *Miscanthus* species using mVISTA, with *S. officinarum* serving as a reference (Figure 8). Although the coding regions were generally conserved across the six *Miscanthus* chloroplast genomes, considerable sequence divergence was found in non-coding regions, particularly between *trnG*^GCC^-*trnG*^UCC^, *atpI*-*atpH*, *atpA*-*rps14*, and *rpl14*-*rps3*. This analysis facilitated a precise examination of sequence variation, revealing minimal variation in chloroplast genes and concentrated variability in non-coding regions.

### 3.5. Phylogenetic Analysis

A maximum likelihood (ML) phylogenetic tree was constructed using 20 complete chloroplast genome sequences, including 13 *Miscanthus* species—comprising nine CWRs, three landraces, and one modern cultivar—along with five *Saccharum* L. species (2 CWRs and 3 modern cultivars) and two *Zea* L. species, to enhance phylogenetic resolution (Figure 9). This analysis was conducted to refine our understanding of relationships within *Miscanthus*, identify closely related crop taxa, and support the integration of target species into the KCWR inventory. The phylogenetic analysis revealed four major clades with high bootstrap value support (96.7/84, 88.6/78, 100/100, and 99.8/99). *M. sinensis* var. *purpurascens*, *M. sinensis*, *M. sinensis* var. *gracillimus*, and *M. sinensis* f. *chejuensis* clustered in Clade 3, with a bootstrap support value of 100/100. Meanwhile, *M.* × *longiberbis* and *M. sacchariflorus* were grouped in Clade 4, supported by a bootstrap value of 99.8/99. A notable finding is that certain *Miscanthus* species formed a distinct clade that was more closely related to the *Saccharum* clade than to Clades 3–4, which comprised the majority of *Miscanthus* species.

## 4. Discussion

In this study, we successfully assembled and analyzed the complete chloroplast genomes of six *Miscanthus* species native to Korea, providing a detailed analysis of structural variation, codon usage patterns, SSR distributions, and phylogenetic relationships. Notably, we identified three *Miscanthus* species (*M. sinensis* var. *gracillimus*, *M. sinensis* f. *chejuensis*, and *M.* × *longiberbis*) that have not yet been registered in the NCBI database. A conserved quadripartite structure—comprising the LSC, SSC, and two IR regions—was observed across all six chloroplast genomes, consistent with the general architecture of land plant chloroplasts. The genome sizes ranged from 141,313 bp to 141,377 bp, with a GC content of 38.4%, which aligns with the typical values reported in other species in the family Poaceae [53,54]. A higher GC content was observed in the IR regions, ranging from 43.9% to 44.1%, compared with that in the LSC and SSC regions. Such enrichment in GC bases is commonly associated with genomic stability. Furthermore, the stable chloroplast genome sizes, which showed only minor variations, indicate a shared evolutionary lineage among these species.

Codon usage analysis of the chloroplast genomes revealed a marked preference for codons ending in A or U, a bias frequently reported in angiosperms [55,56,57]. The preference for A/U–ending codons may enhance the efficiency of gene translation and stabilize the genome by reducing mutations associated with other nucleotide types [58,59]

SSRs—short, repetitive DNA sequences—are widely distributed in the genomes of higher eukaryotes and account or a considerable portion of their non-coding regions [60]. These elements are highly polymorphic and codominant, making them valuable markers in population genetics and phylogenetic research [61,62,63]. In various chloroplast genomes, SSRs are distributed in a non-random manner, predominantly consisting of mononucleotide repeats, with adenine (A) and thymine (T) bases being most common. Long repeat sequences are divided into five groups, ranging from 20 to more than 60 base pairs, as observed in this study. These long repeats may contribute to structural variation within the genome [64,65]. The repetitive sequences discovered in this research may serve as molecular markers for future investigations of the genus *Miscanthus*. These SSRs could be applied to distinguish closely related *Miscanthus* species and to support marker-assisted selection in breeding programs, as demonstrated in related crops such as sugarcane [66,67,68]. Thus, the markers identified expand genomic resources and provide practical tools for conservation and crop improvement.

Previous studies [53,69,70,71] have evaluated the evolution of the chloroplast genome structure in Poaceae, showing that the LSC/IR boundary underwent expansion, causing the *rps19* and *trnH* genes to shift into the IR region. The IRb/SSC junction in the six *Miscanthus* genomes intersected the *ndhF* gene due to its partial duplication, with both *rps19* and *ndhF* consistently occupying IR boundary positions, reflecting the genomic organization observed in other Poaceae species [53,69,70,71]. Although the expansion and contraction of IR regions are common evolutionary events in plants, contributing to genome evolution and structural diversity, the minimal variation observed among the six *Miscanthus* species highlights the remarkable conservation of their chloroplast genomes.

Nucleotide diversity (π) analysis highlighted hotspots of sequence variability in the LSC and SSC portions of the chloroplast genome, whereas the IR regions demonstrated little to no genetic divergence. Coding sequences exhibited substantial conservation across species, whereas non-coding regions displayed markedly higher levels of sequence divergence. Collectively, these results highlight the importance of specific regions as potential genetic markers for evolutionary and phylogenetic studies [72,73,74]. The identified polymorphisms contribute to a deeper understanding of *Miscanthus* chloroplast genome evolution and provide a critical resource for ongoing and future research into the genetic diversity of the genus.

A phylogenetic analysis of 20 chloroplast genomes, including eight *Miscanthus* species, two *Zea* species, and five *Saccharum* species, revealed four major clades with high bootstrap support. These results are consistent with those of previous studies [63], which identified *M. sacchariflorus* and *M. lutarioriparius* as the most closely related species to *M. sinensis* and *M. floridulus.* Together, they constitute a prominent group, determined by molecular markers from chloroplast genome sequences and ITS regions. These markers have demonstrated efficacy in clarifying phylogenetic relationships within the genus. Supporting evidence for the inferred phylogeny of *Miscanthus* was also obtained through cytogenetic investigations using fluorescence and comparative genomic in situ hybridization [75]. Molecular and cytogenetic data collectively contribute to a deeper understanding of the evolutionary history of the genus *Miscanthus*, thereby bolstering the credibility of the reconstructed phylogenetic relationships. Interestingly, certain *Miscanthus* species formed a distinct clade that was more closely related to the *Saccharum* clade than to Clades 3 or 4, suggesting complex evolutionary relationships within the genus. These results align with those of previous studies [13], which reported that *Miscanthus* species originating from eastern and southeastern Asia (*M. sinensis*, *M. sacchariflorus*, and *M. lutarioriparius*) form a monophyletic group.

In contrast, those from Africa and the Himalayas (*M. violaceus*, *M. junceus*, and *M. sorghum*) are more distantly related, indicating biogeographic divergence. This phylogeographic pattern supports the hypothesis that the genus *Miscanthus* has undergone multiple diversification events associated with geographic isolation and other factors. Although African and Himalayan *Miscanthus* species occupy more basal positions in the phylogeny, some genetic similarities with the *Saccharum* lineage have been observed, which may reflect ancient hybridization or reticulate evolution events within the Saccharinae subtribe, as documented in a previous study [76]. Chloroplast data alone cannot fully resolve complex reticulate processes; however, the phylogenetic signals observed in the present study provide indirect evidence of historical gene flow and aid in elucidating broader relationships within the Saccharinae. Future studies incorporating nuclear and mitochondrial data will further refine these evolutionary interpretations [77,78,79].

Furthermore, four *Miscanthus* species, including three landraces and one modern cultivar, were closely related to *M. sacchariflorus*, *M. sinensis* var. *purpurascens*, and *M*. × *longiberbis*. These findings facilitate the identification of closely related crop species and substantially advance efforts to establish the KCWR inventory. Beyond taxonomic clarification, the chloroplast genomes characterized here provide valuable genomic resources that can be directly utilized in future studies. The genomic diversity identified in this study provides a basis for developing molecular markers for stress tolerance and biomass traits [80], which, as noted in previous studies [51,81], demonstrate the utility of chloroplast genome resources in breeding programs and genetic engineering to enhance climate resilience in related crops. In addition to clarifying key aspects of phylogenetic structure and genetic diversity in *Miscanthus*, this study highlights the importance of this genus for breeding strategies aimed at adapting agriculture to environmental challenges.

## 5. Conclusions

This study successfully sequenced and characterized the chloroplast genomes of six *Miscanthus* species native to Korea, expanding genomic resources by including three species not previously registered in public databases. The genomes demonstrated high structural conservation with minimal variation, particularly within coding and IR regions, whereas non-coding regions showed higher variability suitable for genetic marker development. Codon usage bias and SSR distribution patterns were consistent with those observed in other Poaceae members, reflecting evolutionary constraints and genome stability. Phylogenetic analyses revealed four well-supported clades, clarifying the complex evolutionary relationships within *Miscanthus*. Notably, East Asian species exhibited a close sister-group relationship with the *Saccharum* genus, whereas African and Himalayan species formed distinct basal clades. This structure is consistent with previous hypotheses of reticulate evolution in the subtribe Saccharinae. These findings provide critical insights into the taxonomy and genetic diversity of *Miscanthus*, establishing a foundation for future conservation and breeding programs. Although several chloroplast genome studies of Miscanthus have been published previously, Korean taxa remain underrepresented in public genomic databases, have rarely been included in comparative phylogenetic analyses, and largely overlooked in CWR research. In this study, we generated complete chloroplast sequences for six Korean *Miscanthus* species, including three taxa not previously deposited. By incorporating both previously uncharacterized and newly characterized species, this study expands genomic resources, provides unique insights into germplasm, and directly supports the KCWR inventory, thereby contributing to future crop breeding initiatives.

## Figures and Tables

**Figure 1 genes-16-01175-f001:**
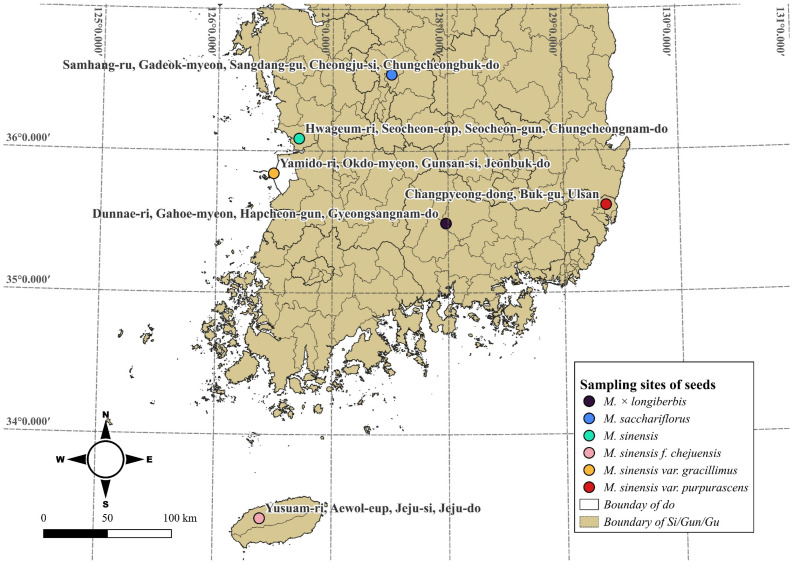
*Miscanthus* seed collection sites. Each species is marked in color, and the collection sites are labeled with brief location descriptions.

**Figure 2 genes-16-01175-f002:**
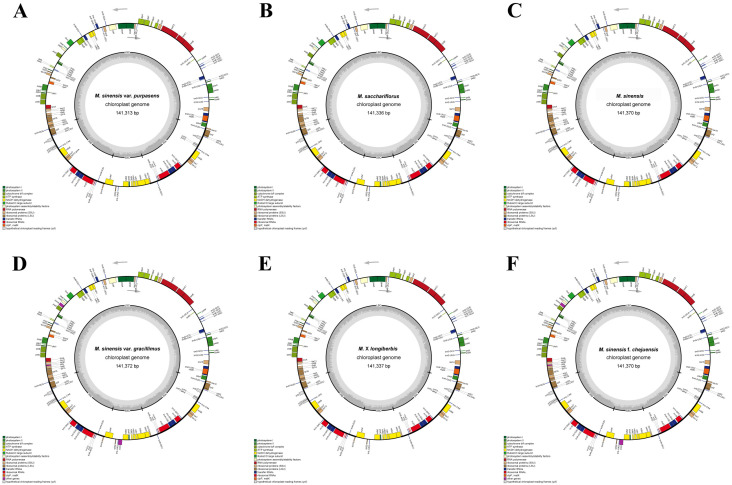
Circular maps of the complete chloroplast genomes of (**A**) *M. sinensis* var. *purpurascens*, (**B**) *M. sacchariflorus*, (**C**) *M. sinensis*, (**D**) *M. sinensis* var. *gracillimus*, (**E**) *M.* × *longiberbis*, and (**F**) *M. sinensis* f. *chejuensis*. Functional gene categories are indicated by distinct color codes. Gene transcription directions are represented by arrows, with those positioned on the outer and inner sides of the circular genome map transcribed in the clockwise and counterclockwise directions, respectively.

**Figure 3 genes-16-01175-f003:**
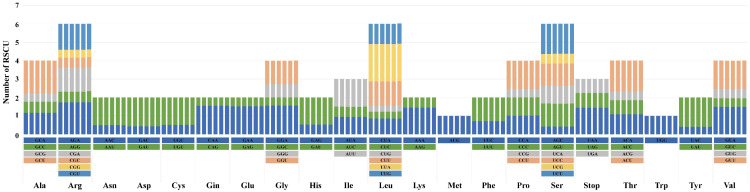
Relative synonymous codon usage (RSCU) patterns for amino acids encoded by the chloroplast protein-coding genes of six *Miscanthus* species. Species are presented from left to right as follows: *M.* × *longiberbis*, *M*. *sacchariflorus*, *M*. *sinensis*, *M. sinensis* f. *chejuensis*, *M. sinensis* var. *gracillimus*, and *M. sinensis* var. *purpurascens*.

**Figure 4 genes-16-01175-f004:**
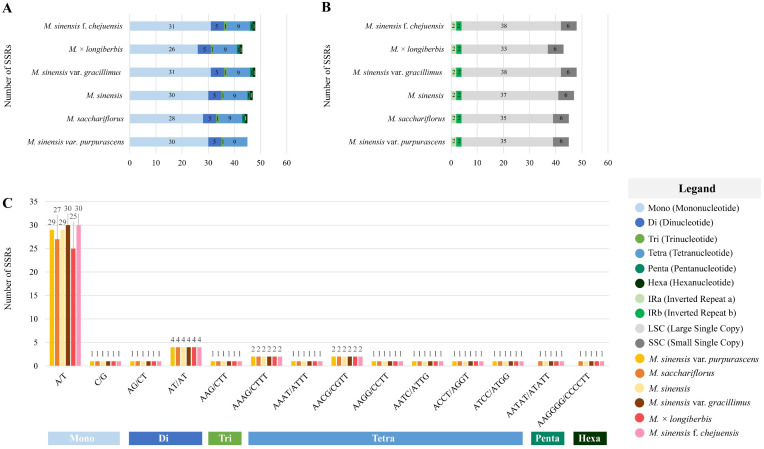
Distribution and classification of simple sequence repeats (SSRs) in the chloroplast genomes of six *Miscanthus* species. The *x*-axis represents SSR types and species, whereas the *y*-axis indicates SSR counts. (**A**) Total number and classification of SSR types. (**B**) Genomic region-specific distribution of SSRs. (**C**) Relative frequency of SSRs categorized by repeat unit type.

**Figure 5 genes-16-01175-f005:**
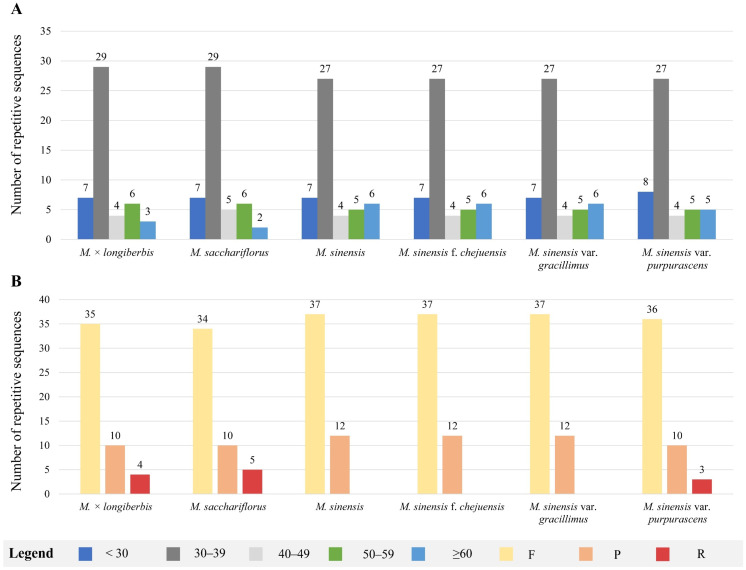
Distribution of long-repeat sequences identified in the chloroplast genomes of six *Miscanthus* species. (**A**) Frequency of repeat types categorized by repeat length. (**B**) Total counts of each repeat type: forward (F), reverse (R), and palindromic (P) repeats.

**Figure 6 genes-16-01175-f006:**
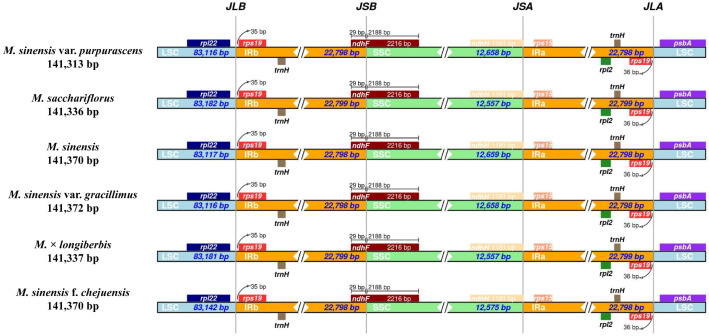
Comparative visualization of junction boundaries between the LSC, SSC, and IR regions in the chloroplast genomes of *Miscanthus* species. Genes adjacent to each boundary are illustrated as boxes, with the distances between genes and boundary regions indicated in base pairs.

**Figure 7 genes-16-01175-f007:**
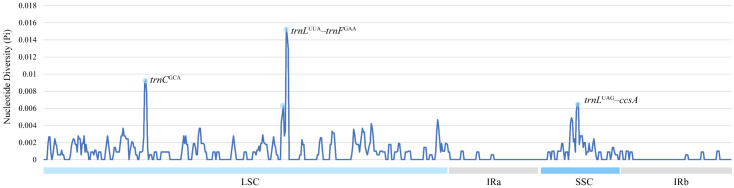
Nucleotide diversity (π) across the chloroplast genomes of six *Miscanthus* species, as revealed by sliding window analysis. Nucleotide diversity is plotted on the *y*-axis, and the *x*-axis reflects the genomic coordinates.

**Figure 8 genes-16-01175-f008:**
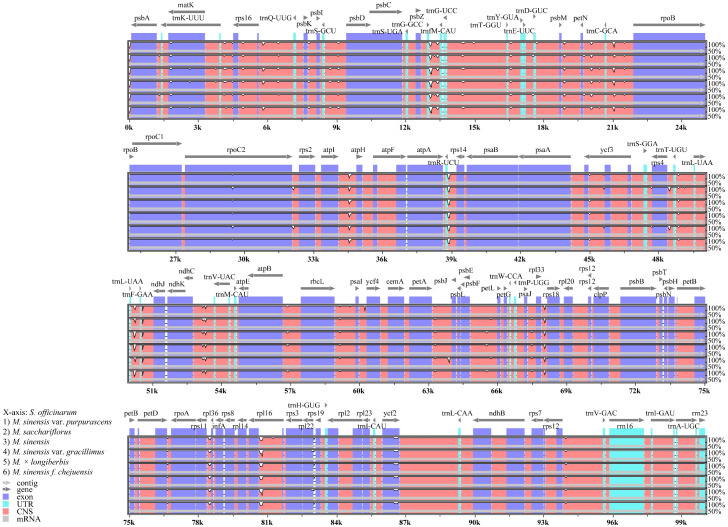
Comparative analysis of chloroplast genomes from six *Miscanthus* species using mVISTA, with *Saccharum officinarum* as the reference. Gray arrows indicate gene orientation and position. Coding regions and conserved non-coding sequences (CNS) are denoted by red, light blue, and blue blocks.

**Figure 9 genes-16-01175-f009:**
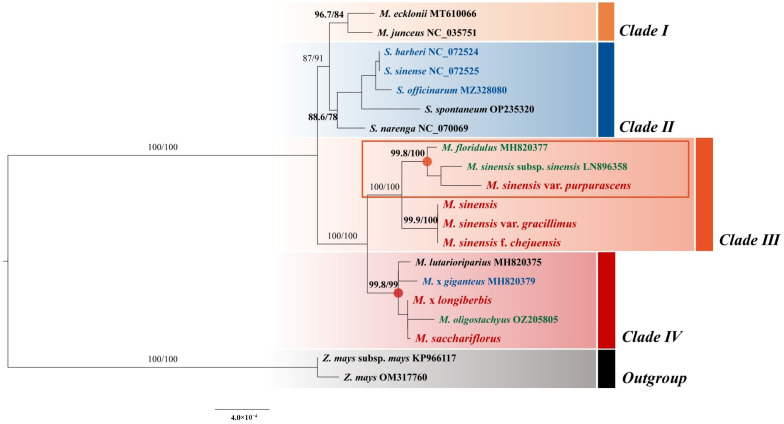
Maximum likelihood phylogenetic tree based on the complete chloroplast genome sequences of *Miscanthus* species. Bootstrap support values (based on 1000 replicates) are shown at each node. Tip labels are color-coded to indicate genetic categories: black for CWRs, green for landraces (local traditional varieties), red for the six *Miscanthus* species analyzed in this study, and blue for modern cultivars, as designated by GRIN–Global. The scale bar represents 0.0004 nucleotide substitutions per site.

**Table 1 genes-16-01175-t001:** List of *Miscanthus* samples and corresponding accession numbers used for chloroplast genome analysis.

No.	Scientific Name	Accession No.	
		Seed	DNA
1	*M. sinensis* var. *purpurascens*	2023-001402	BD002132
2	*M. sacchariflorus*	2023-001419	BD002137
3	*M. sinensis*	2023-001358	BD002141
4	*M. sinensis* var. *gracillimus*	2018-012396	BD002144
5	*M.* × *longiberbis*	2023-001423	BD002147
6	*M. sinensis* f. *chejuensis*	2023-001410	BD002148

**Table 2 genes-16-01175-t002:** Overview of the complete chloroplast genomes of six *Miscanthus* species: (A) *M. sinensis* var. *purpurascens*, (B) *M. sacchariflorus*, (C) *M. sinensis*, (D) *M. sinensis* var. *gracillimus*, (E) *M.* × *longiberbis*, and (F) *M. sinensis* f. *chejuensis*. Values in parentheses represent the number of distinct genes, with redundant copies (e.g., within IR regions) excluded.

Attribute	A	B	C	D	E	F
NCBI	PX334451	PX334449	PX334450	PX334452	PX334453	PX334454
K-BDS	SN01000033	SN01000032	SN01000031	SN01000030	SN01000029	SN01000028
Total length (bp)	141,313	141,336	141,370	141,372	141,377	141,370
LSC length (bp)	83,142	83,181	83,116	83,116	83,117	83,182
IR length (bp)	22,798	22,799	22,798	22,798	22,798	22,799
SSC length (bp)	12,575	12,557	12,658	12,658	12,659	12,557
Total GC content (%)	38.4	38.4	38.4	38.4	38.4	38.4
LSC GC content (%)	36.2	36.2	36.2	36.2	36.2	36.2
SSC GC content (%)	32.8	32.8	32.7	32.7	32.8	32.7
IR GC content (%)	43.9	43.9	44.1	43.9	43.9	43.9
Total genes	135 (113)	135 (113)	135 (113)	135 (113)	135 (113)	135 (113)
CDS genes	89 (79)	89 (79)	89 (79)	89 (79)	89 (79)	89 (79)
tRNA gens	38 (30)	38 (30)	38 (30)	38 (30)	38 (30)	38 (30)
rRNA genes	8 (4)	8 (4)	8 (4)	8 (4)	8 (4)	8 (4)

**Table 3 genes-16-01175-t003:** List of annotated genes in chloroplast genomes of six *Miscanthus* species.

Group of Genes	Gene Symbols
Photosystem I	*psaA*, *psaB*, *psaC*, *psaI*, *psaJ*
Photosystem II	*psbA*, *psbB*, *psbC*, *psbD*, *psbE*, *psbF*, *psbH*, *psbI*, *psbJ*, *psbK*, *psbL*, *psbM*, *psbT*, *psbZ*
Cytochrome b6/f complex	*petA*, *petB*, *petD*, *petG*, *petL*, *petN*
ATP synthase	*atpA*, *atpB*, *atpE*, *atpF*, *atpH*, *atpI*
Rubisco	*rbcL*
NADH dehydrogenase	*ndhA*, *ndhB**, *ndhC*, *ndhD*, *ndhE*, *ndhF*, *ndhG*, *ndhH*, *ndhI*, *ndhJ*, *ndhK*
Proteins of large ribosomal subunits	*rpl14*, *rpl16*, *rpl2**, *rpl20*, *rpl22*, *rpl23**, *rpl32*, *rpl33*, *rpl36*
Proteins of small ribosomal subunits	*rps11*, *rps12*, *rps14*, *rps15**, *rps16*, *rps18*, *rps19**, *rps2*, *rps3*, *rps4*, *rps7**, *rps8*
RNA polymerase	*rpoA*, *rpoB*, *rpoC1*, *rpoC2*
Acetyl–CoA carboxylase	*accD*
C–type cytochrome synthesis gene	*ccsA*
Envelope member protein	*cemA*
Protease	*clpP*
Maturase	*matK*
Translational initiation factor	*infA*
Ribosomal RNAs	*rrn16**, *rrn23**, *rrn4.5**, *rrn5**
Transfer RNAs	*trnA^UGC^**, *trnC^GCA^*, *trnD^GUC^*, *trnE^UUC^*, *trnF^GAA^*, *trnfM^CAU^*, *trnG^GCC^*, *trnG^UCC^*, *trnH^GUG^**, *trnI^CAU^**, *trnI^GAU^**, *trnK^UUU^*, *trnL^CAA^**, *trnL^UAA^*, *trnL^UAG^*, *trnM^CAU^*, *trnN^GUU^**, *trnP^UGG^*, *trnQ^UUG^*, *trnR^ACG^**, *trnR^UCU^*, *trnS^GCU^*, *trnS^GGA^*, *trnS^UGA^*, *trnT^GGU^*, *trnT^UGU^*, *trnV^GAC^**, *trnV^UAC^*, *trnW^CCA^*, *trnY^GUA^*
Hypothetical chloroplast ORFs	*ycf2**, *ycf3*, *ycf4*, *ycf15**, *ycf73**

Gene* = number of copies of a multi-copy gene; ORF = open reading frame.

## Data Availability

All data used in this study are included in the manuscript and have been deposited in NCBI GenBank (https://www.ncbi.nlm.nih.gov/, accessed on 16 September 2025) and K-BDS (https://kbds.re.kr/, accessed on 16 September 2025) under the accession numbers PX334449–PX334454 (NCBI GenBank) and SN01000028–SN01000033 (K-BDS).

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
