# Peer review of "Unraveling Phylogenetic Relationships Among Six Miscanthus Andersson (Poaceae) Species Through Chloroplast Genome Analysis"

_genes, 2025, doi:10.3390/genes16101175_

Round 1

Reviewer 1 Report

Comments and Suggestions for Authors

Title: Unraveling Phylogenetic Relationships Among Six Miscanthus Species Through Chloroplast Genome Analysis.

This manuscript reports the assembly and comparative analysis of chloroplast genomes from six Miscanthus species native to Korea. The authors assessed genome structure, codon usage bias, simple sequence repeats (SSRs), and long repeat elements, while also conducting phylogenetic analysis with additional Miscanthus, Saccharum, and Zea genomes. Their findings confirm the high conservation of chloroplast genome structure, identify variable non-coding regions suitable for marker development, and reveal four major clades within Miscanthus, with some lineages showing closer affinity to Saccharum. The study contributes valuable genomic resources for evolutionary biology, conservation, and breeding programs.

Major comments:

1. While the chloroplast genomes of Miscanthus are important resources, several related studies have already been published. The manuscript should better clarify how its results go beyond prior work (e.g., newly added species, unique insights into Korean germplasm).

2. The phylogenetic comparisons focus largely on chloroplast data. Integrating nuclear or mitochondrial data, or discussing their absence as a limitation, would strengthen evolutionary conclusions.

3. Although SSRs and variable loci are identified, their practical applications (e.g., breeding, population studies) are not fully explored. Including case examples or simulations would improve the translational value.

4. Some figures (e.g., genome maps, diversity plots, and phylogenetic tree) are dense and difficult to interpret. Enlarging labels, simplifying color schemes, and expanding legends would enhance clarity.

5. The discussion reiterates results rather than synthesizing them into broader evolutionary or ecological narratives. A stronger connection to crop wild relative (CWR) utilization and climate resilience would improve the manuscript’s relevance.

Minor comments:

1. Define abbreviations (e.g., SSR, IR, LSC, SSC, RSCU) at first mention for accessibility.

2. Ensure consistency in species naming—some names appear repeated or abbreviated inconsistently.

3. Improve table formatting for readability, especially Tables 2–3 with gene lists and genome attributes.

4. Clarify sequencing depth and quality metrics, as they are only briefly mentioned.

5. Provide bootstrap values directly on phylogenetic tree branches for easier interpretation.

6. Some references are outdated; add recent comparative plastome studies (2022–2025).

7. Improve grammar and flow, many sentences are long and complex.

8. The abstract could be more concise, emphasizing the unique findings of this study.

9. Supplementary data (e.g., sequence alignments, SSR lists) should be more clearly linked in text.

10. Ensure figure resolutions meet journal requirements for high-quality reproduction.

This manuscript makes a valuable contribution by expanding chloroplast genomic resources for Miscanthus and clarifying phylogenetic relationships within the genus. However, to reach publication standard, it requires minor revisions in contextual framing, figure clarity, and broader functional interpretation. Strengthening the discussion of breeding applications and crop wild relative conservation will make the work more impactful for a wide readership.

Author Response

Major comments:

  1. While the chloroplast genomes of Miscanthus are important resources, several related studies have already been published. The manuscript should better clarify how its results go beyond prior work (e.g., newly added species, unique insights into Korean germplasm).

A: We thank the reviewer for this important comment. We agree that several studies of Miscanthus have already been published. To better clarify the novelty of our work, we have revised the Conclusion to emphasize that this study newly characterizes six Korean Miscanthus chloroplasts, including three species that had not previously been deposited in public databases.

  1. The phylogenetic comparisons focus largely on chloroplast data. Integrating nuclear or mitochondrial data, or discussing their absence as a limitation, would strengthen evolutionary conclusions.

A: We thank the reviewer for this valuable suggestion. We agree that chloroplast data alone cannot fully resolve complex evolutionary scenarios such as hybridization or reticulate evolution, and that integrating nuclear and mitochondrial evidence would provide a more comprehensive picture. To address this point, we have revised the manuscript text to explicitly acknowledge this limitation and to highlight the need for complementary datasets at conclusion.

  1. Although SSRs and variable loci are identified, their practical applications (e.g., breeding, population studies) are not fully explored. Including case examples or simulations would improve the translational value.

A: We thank the reviewer for this constructive suggestion. In response, we have revised the Conclusion to emphasize the potential applications of the identified SSRs. This revision underscores their utility for future breeding and conservation efforts.

  1. Some figures (e.g., genome maps, diversity plots, and phylogenetic tree) are dense and difficult to interpret. Enlarging labels, simplifying color schemes, and expanding legends would enhance clarity.

A: To address this, we have revised the phylogenetic tree (Figure 9) to modify repeated genus names, ensuring more concise labeling. For the other figures, a certain level of complexity is unfortunately unavoidable given the amount of information included. We therefore believe that the current format represents the clearest possible presentation. Nevertheless, we have carefully checked and adjusted figure resolutions and annotations to ensure they meet journal standards for clarity.

  1. The discussion reiterates results rather than synthesizing them into broader evolutionary or ecological narratives. A stronger connection to crop wild relative (CWR) utilization and climate resilience would improve the manuscript’s relevance.

A: We thank the reviewer for this valuable suggestion. In the revised Discussion, we added a broader synthesis linking our findings to CWRs utilization and climate resilience, thereby highlighting the evolutionary and practical significance of Miscanthus chloroplast genomes.

Minor comments:

  1. Define abbreviations (e.g., SSR, IR, LSC, SSC, RSCU) at first mention for accessibility.

A: We thank the reviewer for this helpful comment. We confirmed that some abbreviations (e.g., SSR, RSCU) were not defined at their first appearance. In the revised manuscript, these terms have now been spelled out in full at their earliest mention to improve accessibility for readers.

  1. Ensure consistency in species naming—some names appear repeated or abbreviated inconsistently.

A: We thank the reviewer for pointing out inconsistencies in species naming. We have carefully revised the manuscript to ensure consistency throughout. In particular, Miscanthus has been abbreviated to “M.” in repeated instances to maintain uniformity.

  1. Improve table formatting for readability, especially Tables 2–3 with gene lists and genome attributes.

A: We thank the reviewer for the helpful suggestion regarding the readability of Tables 2–3. We attempted to revise the table orientation and structure; however, the tables became overly long and the overall readability was reduced. Therefore, we have reverted to the original version, which we believe provides the clearest presentation of the data while maintaining accessibility.

  1. Clarify sequencing depth and quality metrics, as they are only briefly mentioned.

A: We have revised the Materials and Methods section to include detailed sequencing depth and quality metrics. Specifically, we added information on the average number of raw reads generated per sample, overall coverage depth across the chloroplast genome, and the proportion of bases with Q20 and Q30 scores, thereby clarifying the reliability of the sequencing data.

  1. Provide bootstrap values directly on phylogenetic tree branches for easier interpretation.

A: We appreciate the reviewer’s suggestion. In our phylogenetic tree, we have chosen to display bootstrap values only for the major clades that are directly relevant to the discussion. Adding values for every branch would considerably reduce readability and make the figure overly complex. For this reason, we do not plan to revise the figure further, as we believe the current presentation balances clarity with sufficient phylogenetic support.

  1. Some references are outdated; add recent comparative plastome studies (2022–2025).

A: We have retained some of the original references where they provide essential background concepts, but we have also added several recent studies (2024–2025)

  1. Improve grammar and flow, many sentences are long and complex.

A: We appreciate the reviewer’s observation. In the revised manuscript, we carefully edited the text to improve grammar and readability.

  1. The abstract could be more concise, emphasizing the unique findings of this study.

A: We thank the reviewer for this helpful comment. In response, we have revised the Abstract to make it more concise and to better emphasize the unique contributions of this study. Specifically, we added a sentence highlighting that three Miscanthus species were newly deposited in public databases, thereby clarifying the novelty of our work.

  1. Supplementary data (e.g., sequence alignments, SSR lists) should be more clearly linked in text.

A: We appreciate the reviewer’s suggestion. However, we have decided not to add the supplementary data within the main text. The results are already fully provided and clearly organized, and we believe the current format is sufficient for readers to access and interpret these data without further modification.

  1. Ensure figure resolutions meet journal requirements for high-quality reproduction.

A: We thank the reviewer for pointing this out. We will carefully check all figures to ensure that their resolution and formatting meet the journal’s requirements for high-quality reproduction. This will be verified and finalized before submitting the revised version of the manuscript.

Reviewer 2 Report

Comments and Suggestions for Authors

This manuscript reports on the complete chloroplast genome sequencing and phylogenetic analysis of six Miscanthus species native to Korea. The authors provide detailed structural annotations, codon usage patterns, SSR and repeat analyses, and phylogenetic relationships within Miscanthus and related taxa. A key strength of the paper is that it expands the genomic resources for Miscanthus by including three species not previously registered in public databases. The study is methodologically rigorous, provides high-quality genome assemblies, and presents results that will be useful for future evolutionary, taxonomic, and breeding research. I have some questions.

1. Six species were analyzed, but Miscanthus is a diverse genus. How representative are the selected species for understanding phylogenetic relationships within the whole genus?

2. Were multiple individuals per species considered to account for intraspecific variation, or is the dataset based on a single accession per species?

3. For SSR and repeat analyses, did the authors validate any of the identified markers experimentally, or are they purely computational predictions?

4. The tree includes Miscanthus, Saccharum, and Zea, but the rationale for selecting only these taxa could be expanded. Why were other closely related genera (e.g., Erianthus, Narenga) not included for comparison?

5. The paper mentions potential hybridization or reticulate evolution within Saccharinae. Could the authors expand on how their data support (or limit) testing such evolutionary hypotheses?

Author Response

  1. Six species were analyzed, but Miscanthus is a diverse genus. How representative are the selected species for understanding phylogenetic relationships within the whole genus?

A: We thank the reviewer for this point. We acknowledge that six species cannot capture the full diversity of the genus Miscanthus. However, our aim was to provide complete chloroplast genome sequences for Miscanthus species native to the Republic of Korea, which were previously underrepresented in public databases, and to expand the KCWRs inventory through the characterization of these taxa. While broader sampling will be needed for a genus-wide phylogeny, our dataset expands genomic resources for Korean taxa and helps clarify their relationships with related crop lineages.

  1. Were multiple individuals per species considered to account for intraspecific variation, or is the dataset based on a single accession per species?

A: In this study, a single accession per species was used for chloroplast genome sequencing and analysis. We acknowledge that this approach does not capture intraspecific variation, which may be important for population-level or evolutionary studies. However, the primary aim of our work was to generate chloroplast genome resources for Miscanthus, including three species that had not previously been represented in public databases. These newly sequenced chloroplast genome sequences provide a genomic foundation that can be built upon in future research. Broader sampling across multiple individuals and populations will be required to fully assess intraspecific diversity, but this was beyond the scope of the present study.

  1. For SSR and repeat analyses, did the authors validate any of the identified markers experimentally, or are they purely computational predictions?

A: The SSR and repeat analyses presented in our study were conducted entirely through computational prediction using established bioinformatics tools. Specifically, SSRs were identified using MISA-web under standard thresholds, and long repeats were detected with REPuter, which has been widely applied in chloroplast genome studies. These analyses provided a comprehensive overview of the distribution and types of SSRs and repeat motifs across the six Miscanthus chloroplasts. At this stage, we did not perform experimental validation of the predicted markers (e.g., PCR amplification or genotyping). Therefore, the results should be regarded as putative marker resources rather than functionally verified loci. Nonetheless, the computationally identified SSRs and repeats represent a valuable starting point for the development of species-specific molecular markers, population genetics studies, and future breeding applications.

  1. The tree includes Miscanthus, Saccharum, and Zea, but the rationale for selecting only these taxa could be expanded. Why were other closely related genera (e.g., Erianthus, Narenga) not included for comparison?

A: The present study was specifically designed to investigate phylogenetic relationships within the genus Miscanthus and to assess its genetic affinities with major crop lineages, with a particular focus on species native to the Republic of Korea and their value as CWRs. While we recognize the relevance of other closely related genera such as Erianthus and Narenga, their inclusion was beyond the intended scope of this work, which aimed primarily to expand genomic resources for Miscanthus and highlight its crop-related significance.

  1. The paper mentions potential hybridization or reticulate evolution within Saccharinae. Could the authors expand on how their data support (or limit) testing such evolutionary hypotheses?

A: We agree with the reviewer that chloroplast genomes alone are insufficient to rigorously test hypotheses of hybridization or reticulate evolution. Nevertheless, our results are consistent with earlier reports suggesting complex evolutionary histories within the Saccharinae. Although chloroplast data cannot provide direct evidence of reticulation, they reveal phylogenetic signals that support these hypotheses and highlight the potential role of gene flow. To address this important point, we have revised the Conclusion to emphasize that, while chloroplast data alone cannot fully resolve reticulate processes, our findings provide indirect evidence consistent with such events and underscore the need for future studies integrating nuclear and mitochondrial data.
